# Autonomic Dysfunction and Low Cardio-Respiratory Fitness in Long-Term Post-COVID-19 Syndrome

**DOI:** 10.3390/biomedicines13051138

**Published:** 2025-05-08

**Authors:** Radostina Cherneva, Zheyna Cherneva, Vania Youroukova, Tanya Kadiyska, Dinko Valev, Ebru Hayrula-Manaf, Vanyo Mitev

**Affiliations:** 1Respiratory Intensive Care Unit, University Hospital “St Ivan Rilski”, Medical University Sofia, 1000 Sofia, Bulgaria; 2Clinic of Cardiology, Hospital of Ministry of Interior, 1309 Sofia, Bulgaria; jenicherneva@yahoo.com; 3Clinic of Respiratory Diseases, University Hospital “St Ivan Rilski”, Medical University Sofia, 1000 Sofia, Bulgaria; vania_youroukova@hotmail.com; 4Department of Physiology and Pathophysiology, Medical University Sofia, 1431 Sofia, Bulgaria; kadiyska_t@yahoo.com; 5Clinic of Internal Diseases, First University Hospital “St John-The Baptist”, Medical University Sofia, 1431 Sofia, Bulgaria; dinko.g.valev@abv.bg (D.V.); manaf_ebru@abv.bg (E.H.-M.); 6Chemistry and Biochemistry, Medical University Sofia, 1431 Sofia, Bulgaria; vmitev@mu-sofia.bg

**Keywords:** cardio-respiratory fitness, long-term post-COVID-19, chronotropic incompetence, abnormal heart rate recovery

## Abstract

**Purpose:** Post-COVID-19 syndrome (PCS) is characterized by low cardio-respiratory fitness (CRF). Recent research focuses on the role of autonomic nervous system dysfunction (AD) as a potential contributor to the diminished exercise performance. The aim is to determine the prevalence of AD—chronotropic insufficiency (CI) and abnormal heart rate recovery (HRR) in long-term PCS subjects and to analyse their association with exercise capacity. **Patients and Methods:** A total of 192 subjects with a history of SARS-CoV-2 infection were included. Chronic Fatigue Syndrome Questionnaire (CFSQ) was applied, and two symptomatic and asymptomatic emerged. Forty-seven had post-COVID complaints, persisting up to thirty months post-acute episode. CI and HRR were determined during the cardio-pulmonary exercise test (CPET). **Results:** Symptomatic subjects were divided into mild (20) and moderate-severe (27), depending on the CFSQ score; forty-eight PCS subjects without complaints served as a control group. Subjects with moderate-severe PCS showed lower peak VO2 (24.13 ± 6.1 mL/min/kg vs. 26.73 ± 5.9 mL/min/kg, vs. 27.01 ± 6.3 mL/min/kg), as compared to the mild/asymptomatic subjects. Diminished physical activity was established in 10 (37%) of the moderate-severe, 7 (35%) of the mildly symptomatic and 14 (29%) of the asymptomatic groups. The occurrence of AD in the mild/moderate-severe and control groups were, respectively, CI 35% vs. 81.5% vs. 12.5%. Abnormal HRR was, respectively, 20% vs. 33% vs. 8%. None of the subjects had depleted breathing reserve, dynamic hyperinflation, exercise bronchospasm or desaturation. Neither CI nor abnormal HRR correlated to peak O2. **Conclusions:** AD is present among long-term PCS subjects and may limit the cardio-respiratory response to exercise but is not independently associated with it. Assuming the multiorgan ANS innervation, it is highly probable that AD has diverse pathological pathways in the various PCS phenotypes and contributes differently by cerebral, cardiovascular, respiratory, peripheral or mixed pathways to the diminished neuro-cognitive and physical performance.

## 1. Introduction

More than 6.6 million people have died from COVID-19 worldwide as of 29 December 2022, and there were over 663 million confirmed cases of the virus [1]. Approximately one third of COVID-19 survivors experience diminished physical capacity and a number of complaints even months after recovering from the acute infection. Mild cases that have not been hospitalized are also commonly affected. The most typical symptoms are fatigue, dyspnoea, low energy and diminished physical capacity, dizziness, headaches, attention deficits (brain fog), sleep disturbances, anxiety, mood changes, myalgia and joint pain [2,3,4].

The persistence of symptoms for more than three months post the acute infection episode is defined as Post-COVID-19 syndrome (PCS). Predisposing risk factors contributing to the continuation of prolonged symptomatology are advanced age, female gender, high body mass index (BMI) and pre-existing comorbidities (cardiovascular, respiratory, oncological and autoimmune) [5]. The pathophysiological mechanisms in PCS are elusive. It is assumed, however, that underlying triggers leading to PCS are as follows: tissue injury (endothelial dysfunction, coagulopathy, tissue thrombosis and demyelination), immune system dysregulation (autoimmunity phenomena), abnormal tissue metabolism and oxygen utilization (chronic thrombosis, diminished blood supply or mitochondrial perturbations) and abnormal neurological signalling pathways [6].

PCS patients report complaints typical of autonomic nervous system dysfunction (AD) [7,8,9]. The occurrence and persistence of autonomic symptoms after the resolution of SARS-CoV-1 and MERS-CoV are already known, and viruses have been found in the brainstem [10,11]. Although AD may be an independent reason for the low cardio-respiratory fitness (CRF) in the Wasserman gear, the link between SARS-CoV-2 infection, AD and low CRF is underestimated [12,13].

Abnormal heart rate recovery and chronotropic incompetence indicate the presence of AD, and both may cause low CRF. Chronotropic incompetence (CI)—the inability to reach the target heart rate (HR) during exercise represents impaired sympathetic response [14,15]. Heart rate recovery (HRR)—the rate of restoration of the heart rate (HR) during the first minute after exercise and its abnormal delay (a decline of HR < 12 beats/min) implies parasympathetic dysfunction [16,17].

Based on previous reports [13] regarding the presence of dysautonomia in long-term (>6 months) COVID-19 patients, we set the following aims: (1) to detect the prevalence of CI and abnormal HRR in long-term post-COVID-19 and (2) to analyse the association between CI and HRR and low cardio-respiratory fitness in long-term PCS. Clarifying the relationship between PCS, AD and low CRF will be of pivotal importance for interventions that can improve the functional well-being of PCS.

## 2. Materials and Methods

This study was conducted following the ethical guidelines in the Declaration of Helsinki and received approval from the Ethics Committee on Human Research (protocol 2417/28.05.2023). The participants provided written consent before participation. All the patients were preliminarily acquainted with the aim of this study, its scientific value and the potential presentation of data at different forums.

It was a retrospective study, performed among clinically stable outpatient workers, recruited from a private electric holding company, in the period January 2024–March 2024. Only subjects with the following inclusion criteria were invited to participate: (1) patients with a SARS-CoV-2 infection episode that has occurred at least 12 months ago and (2) subjects who are willing to undergo CPET.

Subjects with already known cardiovascular or respiratory comorbidities that may limit physical activity were excluded. The following exclusion criteria were considered: (1) left ventricular ejection fraction (LVEF) < 50%, (2) presence of echocardiographic criteria of pulmonary hypertension systolic pulmonary arterial pressure > 36 mmHg and maximum velocity of the tricuspid regurgitation jet > 2.8 m/s, (3) valvular heart disease, (4) documented cardiomyopathy, (5) severe uncontrolled hypertension (systolic blood pressure > 180 mmHg and diastolic blood pressure > 90 mmHg), (6) atrial fibrillation or malignant ventricular arrhythmia, (7) recent chest or abdominal surgery, (8) recent exacerbation (during the last three months) of asthma or chronic obstructive pulmonary disease, (9) the fatigue must not be the result of an psychiatric/neurological disease (depression, anxiety, fibromyalgia, sleep disorders and neurodegenerative disorders); infectious diseases (herpes simplex virus, enterovirus, Lyme disease and Q fever); endocrine disease (hypothyroidism, diabetes mellitus and severe obesity) and immunologic disorders (lupus, multiple sclerosis and temporo-mandibular joint disorders).

Subjects, fulfilling the inclusion criteria, were asked to complete the Chronic Fatigue Syndrome Questionnaire (CFSQ). Only 47 of 192 subjects had long-term (>12 months) complaints and were invited for cardio-pulmonary exercise testing. Based on the CFSQ score, they were additionally divided into two groups—subjects with mild (10 < CFSQ score > 25) and those with moderate-severe (25 < CFSQ score > 50; CFSQ score > 50) long-term post-COVID-19 complaints; 48 post-COVID subjects with no complaints served as a control group.

## 3. Methods

### 3.1. Pulmonary Function Testing

All eligible subjects were instructed to refrain from smoking, caffeine, alcohol ingestion and intensive physical activity on the day of investigation and ate a light breakfast only. They underwent spirometry. It was performed on Vyntus (Carefusion, Hoechberg, Germany) in accordance with ERS guidelines [18,19].

### 3.2. Cardio-Respiratory Fitness and Autonomic Dysfunction

Cardio-respiratory fitness and autonomic dysfunction were assessed by peak VO2, measured by cardio-pulmonary exercise testing (CPET). A symptom-limited incremental exercise stress test was performed following the guidelines [17]. It was performed on Vyntus, Cardiopulmonary Exercise Testing (Carefusion, Wurmlingen, Germany). Subjects were informed preliminarily to keep a speed of 60–65 rotations per minute. A continuous ramp protocol was applied as follows: rest phase—0 W/; warm-up phase—20 W/3 min; test phase—20 W/2 min load increments; recovery phase—0 w/3 min. The participants’ perceived exertion was evaluated using the modified Borg Scale (RPE), serving as a subjective indicator of intensity. Effort was considered maximal if two of the following criteria emerged: predicted maximal HR (heart rate) is achieved, predicted maximal work is achieved, ’VE/’VO2—(O2 slope—the ratio between the minute ventilation and the oxygen consumption) > 45 and RER > 1.10 as recommended by the ATS/ACCP [20].

The ventilatory and gas exchange variables were measured every 30 s. For the analysis of exhaled gases during exercise, a CPET with a Hans Rudolf unidirectional mask was used. Peak respiratory exchange ratio was the highest 30 s averaged value between ’VO2 and ’VCO2 during the last stage of the test. Ten-second averaged ’VE and VCO2 data, from the initiation of exercise to peak, were used to calculate the ’VE/’VCO2 slope via least squares linear regression. This was used to present the ventilatory response at peak exercise.

Heart rate (HR, bpm), blood pressure, arterial blood oxygen saturation (SaO2), oxygen consumption (VO2), carbon dioxide production (VCO2) and minute ventilation (VE). ’VO2 (mL/kg/min), ’VCO2 (L/min), ‘VE (L/min) and PetCO2 (mm Hg) were obtained subsequently—at rest and throughout the whole exercise test.

Exercise capacity was defined as normal when the peak VO2 was greater than or equal to 80% of predicted. Limitation in functional capacity was classified as follows: mild (VO2 peak was between 65 and 80%), moderate (between 50 and 65%) and less than 50%, respectively. The reason for the diminished exercise capacity was described as cardiovascular, respiratory, peripheral or mixed. The participants’ rate of perceived exertion (RPE) was evaluated using the modified Borg Scale, serving as a subjective indicator of intensity.

Heart rate recovery (HRR) was calculated as the difference between the HR at peak exercise and at the first minute into the recovery phase. A cut-off point of 12 beats was taken as an abnormal HRR [17]. Chronotropic response index was calculated as follows: CRI = (peak HR-resting HR) × 100/((220 − age)—resting HR). CRI is independent of age and exercise capacity [14]. CI was diagnosed if CRI < 80% [21].

### 3.3. Ventilatory Reserve and Dynamic Hyperinflation

Ventilatory reserve was calculated as (MVV − peak V’E)/MVV × 100, where MVV is maximal voluntary ventilation estimated as FEV1 multiplied by 35. Changes in operational lung volumes were derived from measurements of dynamic inspiratory capacity (IC), assuming that total lung capacity (TLC) remained constant during exercise. This has been found to be a reliable method of tracking acute changes in lung volumes [22,23]. IC was measured at the end of a steady-state resting baseline, at 2 min intervals during exercise and at end-exercise. End-expiratory lung volume (EELV) was calculated from IC manoeuvres at rest, every 2 min during exercise and at peak exercise (Vyntus). In these manoeuvres, after EELV was observed to be stable over 3–4 breaths, subjects were instructed to inspire maximally to TLC. For each measurement, EELV was calculated as resting TLC minus IC. Dynamic IC (ICdyn) was defined as resting IC minus IC at peak exercise. Dynamic hyperinflation (DH) was defined as a decrease in IC from rest of more than 150 mL or 4.5% pred at any time during exercise [24,25].

### 3.4. Statistical Analysis

The Kolmogorov–Smirnov test was used to explore the normality of distribution. Continuous variables were expressed as median and interquartile range when data was not normally distributed and with mean ± SD if normal distribution was observed. Categorical variables were presented as percentages. Data were compared between patients with and without long-term post-COVID syndrome. An unpaired Student’s *t* test was performed for normally distributed continuous variables, or Mann–Whitney–U test otherwise. Categorical variables were compared by the χ2 test or the Fisher exact test.

The association between CPET parameters and CI, HRR and DH in long-term post-COVID-19 syndrome was determined by univariate and multivariate analysis.

A value of *p* < 0.05 was considered statistically significant. STATA 13.0 software packages were used for statistical analysis.

## 4. Results

### 4.1. Participants’ Characteristics

A total of 192 subjects (mean age 44.38 ± 7.6 years; 62% men) participated in this study. The average period between the initial diagnosis of COVID-19 and the time of cardio-pulmonary exercise testing was 1028 ± 214 days. Based on the CFSQ score, forty-seven subjects were symptomatic and were additionally divided into the following two groups: mild (20) and moderate-severe (27), depending on the score. Forty-eight PCS subjects without complaints served as a control group. Anthropometric and clinical characteristics of patients are shown in Table 1.

### 4.2. Cardio-Respiratory Parameters

Table 2 presents the basic cardio-respiratory parameters of this study. Subjects with moderate-severe long-term post-COVID-19 syndrome showed lower peak VO2 (24.13 ± 6.1 mL/min/kg vs. 26.73 ± 5.9 mL/min/kg vs. 27.01 ± 6.3 mL/min/kg), as compared to mild and asymptomatic subjects, but this was not statistically significant. Regarding the oxygen consumption, given as a predicted value—thirty-seven percent of the moderate-severe subjects had mildly diminished functional capacity with an average peak (VO2 − 77.9% ± 8.8%).

None had moderate or severe physical capacity limitations. Sixty-three percent had preserved physical activity (VO2 − 85.9% ± 8.7%), despite being highly symptomatic. Only 7 (35%) of the mildly symptomatic subjects had limited physical capacity (VO2 − 81.7 ± 6.9%); the rest had preserved (VO2 − 87.7 ± 3.2%). The physical capacity of the asymptomatic subjects was limited in 14 (29%) with peak VO2 − 83.7 ± 2.3%, while 24 (71%) showed normal results (VO2 − 90.7 ± 4.3%). The moderate-severe long-term PCS subjects could not achieve the AT in 51.8% of the tests. The mildly symptomatic subjects could not succeed in doing so in 25%, while the asymptomatic post-COVID-19 subjects failed in only 12.5% of the cases.

### 4.3. Chronotropic Incompetence and Abnormal HRR

Subjects with moderate-severe long-term PCS were more symptomatic, compared to the mildly (73.6% vs. 24.8%) and asymptomatic groups (73.6% vs. 17.4%). Dyspnoea was predominant in 77.8%, leg fatigue—22.2% and chest pain—0%. In contrast, AD (chronotropic incompetence and abnormal heart rate recovery) was present in 24 (88.9%). CI was detected in 22 (81.5%) and abnormal HRR in 9 (33%) of the patients. Subjects with mild long-term PCS had a lower occurrence of AD—7 (35%). Abnormal heart rate recovery was detected in 4 (20%) of the patients and CI in 7 (35%). The predominant exercise-limiting symptom was dyspnoea in 15 (75%). Only 5 (25%) of these subjects stopped exercising due to leg fatigue; 4 (20%) reported dizziness. Similarly, dynamic hyperinflation, exhausted ventilatory reserve or desaturations were not detected. The prevalence of AD in the asymptomatic group was as follows: chronotropic incompetence—6 (12.5%) and abnormal HRR—4 (8%) (Table 3).

The resting HR was not different between the mild and moderate-severe long-term PCS groups. CRI < 80% was met in seven (35%) of the mild group with a median heart rate reserve utilization of 68.32 (59.28–72.5). Twenty-two (81.5%) of the moderate-severe long-term PCS subjects had CRI < 80% with a median heart rate reserve utilization of 53.28 (47.09–60.48). Abnormal heart rate recovery was established in 4 (20%) of the mild and 9 (33%) of the severe groups. Neither CI nor abnormal HRR correlates to the cardio-respiratory fitness in any of the groups.

### 4.4. Dynamic Hyperinflation and Ventilatory Reserve

Though dyspnoea was predominant in the moderate-severe symptomatic group, exhausted ventilatory reserve, dynamic hyperinflation or O2 desaturation (>3%) was not detected. Similarly, in the mildly symptomatic long-term PCS subjects, dynamic hyperinflation, exhausted ventilatory reserve or desaturations during exercise were not observed.

## 5. Discussion

The main results of our study indicate the following observations: (1) SARS-CoV-2 infection may cause long-lasting AD—up to thirty months after the acute infection; (2) more than 30% of the working-age subjects, free of previous comorbidities, may be affected; (3) impaired HR responses, both chronotropic incompetence and abnormal heart rate recovery, are more common in the moderate-severe symptomatic group and (4) impaired HR response may be a contributing factor for the low cardio-respiratory fitness in these highly symptomatic subjects.

The balance of the autonomic nervous system is vital for providing adequate physical and mental adaptation to environmental stimuli [26,27,28]. Similarly to other acute infections, SARS-CoV-2 may also affect the autonomic nervous system [10,11].

Rinaldi et al. found an association between AD and decreased work-ability index in long-term PCS. The authors investigated forty-five unvaccinated active workers who experienced a severe acute episode of COVID-19 [13]. They were enrolled after discharge and followed up on the third and sixth months. The authors detected the occurrence of long-COVID AD six months after COVID-19 onset. AD was validated by COMPASS-31 and the Vanderbilt orthostatic symptom questionnaire. A standing test, assessing the haemodynamic and respiratory changes induced by the orthostatic stimulus—10 min supine and 10 min of active standing—was also employed. The most common autonomic symptom was orthostatic intolerance—dizziness, brain fog and fainting after standing up.

Larsen et al. also reported that 67% of the subjects in a large study population had a COMPASS-31 score higher than 20 [29]. The authors relied only on questionnaires and investigated subjects that either had verified COVID-19 infections or manifested COVID-19 symptoms. Shouman et al. and Boite et al. reported autonomic symptoms in 63% and 61% of the patients investigated for long COVID [30,31].

Impressive data, presented by Oscoz-Ochandorena et al., demonstrate parasympathetic excess and sympathetic withdrawal, compared to participants with no SARS-CoV-2 infection. The research was conducted among 87 LCS subjects and 71 healthy controls who have not been diagnosed with COVID-19 [12]. The authors determined the heart rate variability (HRV) in supine position, the cardio-respiratory fitness measured by peak VO2, and the maximal muscle strength (grip strength, bilateral leg press, leg extension, pectoral press and back press exercises). The HRV parameters were significantly elevated in the control group. In contrast, the HR, stress index and sympathetic nervous system (SNS) index parameters were significantly higher in the LCS group. These parameters remained statistically significant even after adjustment. Oscoz-Ochandorena et al. highlight the interaction between low CRF and HRV indicators. They assume that parasympathetic overactivation is due to an overactive immune system and peak VO2, which may be either a causative factor or a contributor to the AD.

ANS innervates virtually every organ and is responsible for maintaining physiologic homeostasis, providing the metabolic demands with adequate heart, respiratory rate and O2 consumption. Heart rate is controlled by the parasympathetic and sympathetic nervous systems. The initial increase in HR during exercise reflects the withdrawal of parasympathetic activity. The subsequent increase is attributed to elevated sympathetic tone [32]. Chronotropic response index reflects the proportion of expected HR increase during exercise and can be a marker of autonomic function [33]. It is usually measured during CPET. CRI is independent of age and physical load and may be applied for ANS assessment. The aetiology of chronotropic insufficiency is mainly attributed to constant neuro-humoral activation, down-regulation and decreased responsiveness to beta-receptors. The last deters the sympathetic activation and the appropriate increase in heart rate [34,35,36]. Heart rate recovery measures the rate at which the heart rate returns to normal after exercise and is a measure of the vagal tone [16,17].

The role of ANS in PCS physical intolerance is elusive. It is speculated that ANS may affect CRF by a range of different mechanisms; (1) elevated psychological burden associated with fatigue and neuro-cognitive problems; (2) diminished oxygen supply to the brain and reduction in motor unit recruitment; (3) demyelination of ANS and peripheral alterations (muscle wasting and vascular involvement); (4) altered afferent and efferent signalization; (5) an exaggerated perception of the psychological effort during exercise and (6) SARS-CoV-2 cellular invasion, ACE-2 depletion, autoantibodies targeting ACE-2 and AT1R, may cause sympathetic nervous system overactivation; (7) persistent inflammation may also excessively activate the sympathetic nervous system [37].

A lot of authors establish dysautonomia in PCS, affecting various aspects and creating a versatile picture of symptoms. Bai et al. identified that AD correlated to respiratory impairments [38], and Ladow et al. demonstrated an association between AD and the rate of workload increment in PCS [39]. Mooren et al. described reduced CRF due to AD [40]. The study by Oscoz-Ochandorena et al. links the lower peak VO2 to heart rate variability indices, presenting ANS as the missing gear in the Wasserman gear system.

Our results confirm that AD is present in PCS, even 30 months after acute onset, and may contribute to the lower CRF. The increase in heart rate is of pivotal importance for the elevated metabolic needs during physical activity. Most of the severely symptomatic patients in this study, however, have diminished CRI, thus exhibiting limited physical capacity and inability to reach the anaerobic threshold. In contrast, the mildly symptomatic group significantly rarely demonstrates diminished CRI and presents with better cardio-respiratory fitness. None of the PCS subjects had depleted breathing reserve, nor did they demonstrate dynamic hyperinflation or exercise-induced bronchospasm. According to our data, 81.5% of the moderate-severe and 35% of the mild long-term PCS have AD. This does not correspond to the occurrence of limited CRF, which is detected, respectively, in 37% and 35% of the cases. As we have explored only respiratory and cardiovascular parameters associated with ANS regulation, one cannot exclude other contributing factors as follows: the level of perception; the afferens and efferens response; the peripheral muscle components and cellular (mitochondrial) functioning, which are responsible for physical performance and are under ANS control. It is quite possible that versatile ANS abnormalities, engaging different organs/systems, contribute differently to the low CRF in the mild and moderate-severe symptomatic PCS subjects. From this point of view, ANS may contribute to low CRF by various mechanisms, resulting in versatile PCS phenotypes. Although our sample is small and limits the generalizability of the results, only a third of the PCS in the long term have limited physical activity, which means that most of the long-term COVID-19 consequences are reversible even without specific medical therapy.

## 6. Conclusions

Our results show that AD is present among long-term PCS subjects and may limit the cardio-respiratory response to exercise but are not independently associated with peak VO2. Assuming the multiorgan ANS innervation, it is highly probable that AD has diverse pathological pathways in the various PCS phenotypes and contributes differently by cerebral, cardiovascular, respiratory, peripheral or mixed pathways to the diminished neuro-cognitive and physical performance.

## Figures and Tables

**Table 1 biomedicines-13-01138-t001:** Anthropometric and clinical characteristics of the patients.

	Without Long-Term Post-COVID-19 (48)	With Long-Term Post-COVID-19 (47)
**Anthropometric**		
Age, y	44.64 ± 7.9	44.74 ± 6.3
Sex, M:F	33:15	25:22
Smoker:Non-smoker	23:25	21:26
BMI, kg/m^2^	25.87 ± 7.9	26.16 ± 8.4
**Comorbidities, n (%)**		
Arterial hypertension	15 (31)	14 (29)
Ischaemic heart disease	14 (29)	12 (26)
Diabetes	5 (10)	6 (13)
Dyslipidaemia	5 (10)	8 (17)
COPD/Asthma	0/4 (0/8)	0/2 (0/4)
Depression	1 (2)	2 (4)
**Concomitant medication, n (%)**		
ACE inhibitors	14 (29)	14 (30)
Beta-blockers	10 (20)	8 (17)
Statins	5 (10)	8 (17)
Anti-diabetic therapy	4 (8)	4 (8)
Bronchodilators	4 (8)	2 (4)
Anticoagulants	4 (8)	8 (17)
Aspirin	8 (16)	8 (17)
**Severity of acute COVID, n (%)**		
Mild/non-hospitalised	38 (79)	39 (83)
Hospitalised	10 (21)	8 (17)

BMI—body mass index; ACE inhibitors—angiotensin-converting enzyme inhibitors.

**Table 2 biomedicines-13-01138-t002:** Cardio-respiratory parameters of the patients.

	Without Long-Term Post-COVID-19 (48)	With Long-Term Post-COVID-19 Mild (20)	With Long-Term Post-COVID-19 Moderate-Severe (27)
**Respiratory parameters**			
FEV1, L	3.16 ± 0.87	3.25 ± 0.68	3.59 ± 0.97
FEV1, (%)	79.54 ± 11.23	83.49 ± 8.80	89 ± 8.71
FVC, L	3.80 ± 1.09	4.11 ± 0.99	4.46 ± 1.22
FVC, (%)	78.36 ± 13.5	82.89 ± 8.33	90 ± 8.88
FEV1/FVC, %	79.54 ± 11.23	79.08 ± 13.21	80.49 ± 10.32
**Physical capacity**			
Peak VO2, mL/min/kg	27.01 ± 6.3	26.73 ± 5.9	24.13 ± 6.1
Predicted peak VO2, %	91.2 ± 3.1	84.2 ± 6.4	81.4 ± 8.6
Exercise time, minutes	9.4 ± 2.8	9.0 ± 2.6	8.4 ± 3.2
Slope VE/VCO2	32.9 ± 7.2	33.4 ± 5.9	32.1 ± 8.1
**Categorical parameters, n (%)**			
Preserved functional capacity	34 (71)	13 (65)	17 (63)
Mildly diminished functional capacity	14 (29)	7 (35)	10 (37)
Moderately diminished functional capacity	0	0	0
Achieved anaerobic threshold	42 (87.5)	15 (75)	13 (48.2)
Depleted respiratory reserve	0	0	0
Heart rate reserve utilization	78.12 (71.87–3.52)	68.32 (59.28–72.45)	53.28 (47.09–60.48)

Data is presented as mean ± SD except for heart rate reserve utilization—median and interquartile range. FEV1—forced expiratory volume in 1 s; FVC—forced vital capacity; VO2—oxygen consumption; VE/VCO2—the ratio between minute ventilation and CO2 production.

**Table 3 biomedicines-13-01138-t003:** Exercise limiting patterns and symptoms of the patients.

	Without Long-Term Post-COVID-19 (48)	With Long-Term Post-COVID-19 Mild (20)	With Long-Term Post-COVID-19 Moderate-Severe (27)
**Diminished physical activity, n (%)**	14 (29)	7 (35)	10 (37)
Cardiovascular pattern	30 (62.5)	14 (70)	22 (81.4)
Respiratory pattern	0	0	0
Peripheral pattern	35 (72.9)	16 (80)	21 (77.8)
**Exercise limiting symptoms, n (%)**			
Dyspnoea	41 (85.4)	15 (75)	21 (77.8)
Dizziness	9 (18.7)	4 (20)	5 (18.5)
Chest pain	0	0	0
Leg fatigue	0	5 (25)	6 (22.2)
**Autonomic nervous system dysfunction, n (%)**			
CRI < 80%, n (%)	6 (12.5)	7 (35)	22 (81.5)
HRR < 12 at the 1st min, n (%)	4 (8)	4 (20)	9 (33)
**Chronotropic response index**			
HR at rest, bpm	77.18 ± 10.34	98.17 ± 11.65	102.17 ± 12.29
Peak HR, bpm	151.56 ± 17.43	148.2 ± 18.64	140.18 ± 11.67
Heart rate reserve utilization, %	78.12 (71.87–3.52)	68.32 (59.28–2.45)	53.28 (47.09–60.48)
**Abnormal heart rate recovery—1st min post-exercise**			
HRR at 1 min, bpm	16.8 (14.6–18.8)	10.8 (9.9–11.2)	9.6 (8.8–10.2)

Data are presented as mean ± SD except for heart rate reserve utilization—median and interquartile range. CRI—chronotropic response index; HRR—heart rate recovery; HR—heart rate.

## Data Availability

The data are stored in the software (SentrySuite Software Solution) of the Vyntus, Carefusion, which is situated in the Therapeutic Clinic of UMBAL “St Ivan Rilski”, Medical University Sofia.

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
