# Peer review of "Autonomic Dysfunction and Low Cardio-Respiratory Fitness in Long-Term Post-COVID-19 Syndrome"

_biomedicines, 2025, doi:10.3390/biomedicines13051138_

Round 1
Reviewer 1 Report
Comments and Suggestions for Authors
The manuscript entitled 'Autonomic dysfunction and low cardio-respiratory fitness in long-term post-COVID-19 syndrome' has scope in the specified field but needed some corrections in some parts of the manuscript.
Comments:
- No figures/Charts/Graphs are included in the manuscript. To capture the attention of the viewer, Figures are important and deliver complex scientific information with a clarity that plain text alone cannot achieve.
- Abbreviations were given in the text and also suggested to mention below the tables.
- In numerical, use point properly in text and tables (used comma in place of point)
Author Response
- No figures/Charts/Graphs are included in the manuscript. To capture the attention of the viewer, Figures are important and deliver complex scientific information with a clarity that plain text alone cannot achieve.
Response 1: Dear reviewer, I totally agree that charts, figures and graphs make the text more comprehensive to the reader. Unfortunately, we did not find any statistical correlations that may be presented by graphs or charts. The data that is obtained by the CPET may be presented with the nine panel plot of Wasserman, but it is specific for each of the subjects, so we cannot make a summary for the whole group. This is the main reason for the lack of the above mentioned.
2.Abbreviations were given in the text and also suggested to mention below the tables.
Response 2. Dear reviewer, I have mentioned the abbreviations in the text, as well as, added them in the tables below.
3.In numerical, use point properly in text and tables (used comma in place of point)
Response 3. Dear reviewer, you are right, regarding the use of point. I have corrected it throughout the whole text, as well as, in the tables.
Reviewer 2 Report
Comments and Suggestions for Authors
A review report of the manuscript entitled “Autonomic Dysfunction and Low Cardio-Respiratory Fitness in Long-Term Post-COVID-19 Syndrome”
- Authors are suggested to proofread the manuscript after addressing all comments to avoid any typological, grammatical, and lingual mistakes and errors. For example, the term “SARSCoV-2” on line 29 which should be “SARS-CoV-2”, and many more throughout the text.
- (lines 53-55) To support the first sentence in the introduction, I suggest that the authors give a bit of context on the number of COVID-19 cases happened globally. For example, “More than 6.6 million people have died from COVID-19 worldwide as of December 29, 2022, and there were over 663 million confirmed cases of the virus.” This can be cited from https://narrax.org/main/article/view/71/49
- (lines 55-58) This sentence “The most typical symptoms are fatigue, dyspnea, low energy and diminished physical capacity, dizziness, headaches, attention deficits (brain fog), sleep disturbances, anxiety, mood changes, myalgia, joint pain” also shares the same idea with a similar study by Fajar et al. https://pmc.ncbi.nlm.nih.gov/articles/PMC10914045/ Kindly include this reference to make the sentence stronger.
- (lines 129-130) "Subjects were informed pre- liminary to keep a speed of 60-65 rotations per minute." "Preliminary" is used incorrectly here; it should be "preliminarily."
- (lines 134-135) Clarify the units and terms like "HR" and "VE/VO2" for readers unfamiliar with the abbreviations. The authors should introduce those terms in their first mention. The current manuscript introduced them quite late in the text, which is not very effective, I would say.
Author Response
- Authors are suggested to proofread the manuscript after addressing all comments to avoid any typological, grammatical, and lingual mistakes and errors. For example, the term “SARSCoV-2” on line 29 which should be “SARS-CoV-2”, and many more throughout the text.
Response.1. Dear reviewer, I totally agree that there are a lot of typological mistakes. I have read through the whole manuscript, and I think, I have corrected most of them as you can see.
- (lines 53-55) To support the first sentence in the introduction, I suggest that the authors give a bit of context on the number of COVID-19 cases happened globally. For example, “More than 6.6 million people have died from COVID-19 worldwide as of December 29, 2022, and there were over 663 million confirmed cases of the virus.” This can be cited from https://narrax.org/main/article/view/71/49
Response.2. Dear reviewer, I have cited the numbers and added the recommended reference to the paper.
- (lines 55-58) This sentence “The most typical symptoms are fatigue, dyspnea, low energy and diminished physical capacity, dizziness, headaches, attention deficits (brain fog), sleep disturbances, anxiety, mood changes, myalgia, joint pain” also shares the same idea with a similar study by Fajar et al. https://pmc.ncbi.nlm.nih.gov/articles/PMC10914045/ Kindly include this reference to make the sentence stronger.
Response.3. Dear reviewer, after reading the work by Fajar et al, I totally agree with you and have also added this reference to the manuscript.
- (lines 129-130) "Subjects were informed pre- liminary to keep a speed of 60-65 rotations per minute." "Preliminary" is used incorrectly here; it should be "preliminarily."
Response.4. Dear reviewer, this of course is a technical mistake, which is corrected.
- (lines 134-135) Clarify the units and terms like "HR" and "VE/VO2" for readers unfamiliar with the abbreviations. The authors should introduce those terms in their first mention. The current manuscript introduced them quite late in the text, which is not very effective, I would say.
Response.5. Dear reviewer, I totally agree with you and have taken into consideration the remark, as you can check in the body text.
Reviewer 3 Report
Comments and Suggestions for Authors
In their study, Cherneva et al. perform spiroergometry tests on patients who have recovered from COVID-19. Some of the study collective reported persistent symptoms in the long-term follow-up after the infection (mainly dyspnea; the symptomatic group was divided into 'mildly symptomatic' and 'moderately to severely symptomatic' based on the results of a questionnaire). Asymptomatic post-COVID patients served as a control group. The authors were able to show that the more symptomatic patients had only a slight (insignificant) reduction in maximum oxygen uptake, but there were clear signs of autonomic dysfunction. For example, symptomatic post-COVID patients showed a higher degree of abnormal chronotropic response index and poorer heart rate reserve utilization. With their work, the authors contribute important information to the phenomenological description of the heterogeneous construct “post-covid-syndrome”. In this context, (postural) tachycardia syndromes and other indications of autonomic dysregulation have already been described, which is confirmed in the present study collective. The following points should be explicitly addressed:
- The authors describe the pathological conditions that represented exclusion criteria for the study collective as potential confounders. It would be of great interest to learn more about how it was ensured that the exclusion criteria were actually excluded.
- The authors should discuss in detail how the observed phenomena (regarding the exercise-dependent heart rate response) are generally associated with the training status of the subjects examined. For example, a (however caused) subjective limitation of performance capacity may be associated with a poor training status, which is reflected in the CPET as autonomic dysregulation (or deconditioning) In general, the well-known question of causality and correlation arises.
- In some studies in which CPET was performed in post-COVID patients, chaotic breathing patterns were observed that were associated with early hyperventilation. Were such phenomena also observed in this study group?
Author Response
The authors describe the pathological conditions that represented exclusion criteria for the study collective as potential confounders. It would be of great interest to learn more about how it was ensured that the exclusion criteria were actually excluded.
- Response 1: Dear reviewer, I fully agree that it seems difficult to consider all the exclusion criteria, but as it is mentioned in the text the subjects are recruited from a private electro-company. They all have good medical archive as they receive additional healthcare insurance. So, the medical history of all of them is nicely archived and all the annual preventive examinations, hospitalizations, primary care consultations, laboratory or imaging tests are good organized and stored. The collection regarding their medical history, concomitant medication and lab and instrumental tests was easy to obtain and analyse.
- The authors should discuss in detail how the observed phenomena (regarding the exercise-dependent heart rate response) are generally associated with the training status of the subjects examined. For example, a (however caused) subjective limitation of performance capacity may be associated with a poor training status, which is reflected in the CPET as autonomic dysregulation (or deconditioning) In general, the well-known question of causality and correlation arises. - Response 2: Dear reviewer, I think that there is some misunderstanding.We all know that deconditioned patients hardly reach a submaximal or maximal threshold of exercise and besides all, the curves on the 9-panel plot of Wasserman are very typical and easily distinguished from cardiac, peripheral or respiratory patterns for termination of the test. As all of the patients that have been included in the analysis have fulfilled the criteria for maximal load we cannot speak for physical inactivity as e reason for stopping the test.
In some studies in which CPET was performed in post-COVID patients, chaotic breathing patterns were observed that were associated with early hyperventilation. Were such phenomena also observed in this study group? - Response 3: Dear reviewer, we also had chaotic breathing, but I have not mentioned it in the study because the topic is autonomic dysfunction with cardiac responses being the featured parameters.
Round 2
Reviewer 2 Report
Comments and Suggestions for Authors
I commend the authors for addressing my concerns. Now this manuscript has improved a lot and is ready for further consideration.
Reviewer 3 Report
Comments and Suggestions for Authors
The manuscript has substantially improved after incorporating some new information.